# Effects of Gestational Sleep Patterns and Their Changes on Maternal Glycemia and Offspring Physical Growth in Early Life

**DOI:** 10.3390/nu14163390

**Published:** 2022-08-18

**Authors:** Jiaojiao Zou, Qian Wei, Peiqi Ye, Yuyang Shi, Yunhui Zhang, Huijing Shi

**Affiliations:** 1Department of Maternal, Child and Adolescent Health, School of Public Health, Fudan University, No. 130 Dong’an Road, Shanghai 200032, China; 2Department of Environmental Health, School of Public Health, Fudan University, No. 130 Dong’an Road, Shanghai 200032, China

**Keywords:** gestational sleep patterns, gestational diabetes mellitus (GDM), offspring physical growth, cohort study

## Abstract

Maternal sleep patterns during pregnancy are drawing increased attention to examine its role in the regulation of maternal glycemia and physical growth of offspring within 24 months. Among 3329 eligible mother–child pairs included in the Shanghai Maternal–Child Pairs Cohort, sleep patterns of pregnant women were assessed by Pittsburgh Sleep Quality Index and objective measurement in early and late pregnancy. Offspring physical growth within 24 months was primarily indicated by the body mass index Z-score (BAZ), catch-up growth, and overweight/obesity. In total, 3329 and 382 pregnant women were included with subjectively assessed and objectively measured sleep pattern, respectively. The increased risk of GDM was associated with maternal night-time sleep duration ≥8.5 h in early pregnancy, or sleep quality change from poor to good during pregnancy (OR = 1.48; 95% CI, 1.06 to 2.07). In the GDM group, the effect of sleep duration in early pregnancy on overweight/obesity in offspring within 24 months showed a U-shaped curve, with a 1.73-fold and 1.43-fold increased risk of overweight/obesity of offspring in pregnant women with <7.5 or ≥8.5 h of sleep duration, respectively. A good gestational sleep pattern was required to reduce the risk of GDM and offspring overweight/obesity within 24 months.

## 1. Introduction

Hyperglycemia during pregnancy has adverse effects on offspring growth, such as an increased risk of overweight/obesity in early life [1]. As obesity persisted from birth to two years of age and even into adulthood [2,3], according to the Developmental Origins of Health and Disease (DOHaD) [4], early pregnancy planning should be incorporated as a preventable factor in child obesity interventions. Early pregnancy placental function and gene expression are influenced by a women’s metabolic status, and placental dysfunction is most likely to occur in early pregnancy, before most pregnancy intervention trials have begun. Lifestyle interventions initiated after early pregnancy had less impact on placental gene expression and reduced perinatal morbidity [5]. Pregnancy is an important period to encourage lifestyle changes to prevent maternal hyperglycemia, which usually include exercise and dietary factors. Sleep factors can affect normal glucose levels and be modified during pregnancy [6], but have been easily overlooked.

Typical sleep patterns are described by sleep quality and sleep duration. Primiparas have a difficult time returning to their pre-pregnancy sleep pattern after delivery, as their sleep patterns were altered in early pregnancy [7]. In late pregnancy, women awakened more often and slept for 25–40 min less at night compared to early pregnancy, and 87% of pregnant women experienced sleep problems [8]. Studies on non-pregnant women have suggested that the sleep duration impaired insulin sensitivity and glucose metabolism, increasing the risk of type II diabetes [9,10]. Two existing meta-analyses have reached conflicting conclusions as to whether the sleep duration during pregnancy affects the risk of developing GDM [11,12]. A large cohort study concluded that pregnant women with poorer sleep quality were more likely to develop GDM regardless of sleep duration [13]; however, it remains controversial whether decreased sleep quality across weeks of gestation has an impact on glycemia. Therefore, it was important to conduct a longitudinal assessment of sleep patterns and their changes in different stages of pregnancy on glucose levels during pregnancy.

Alterations in sleep patterns have been shown to increase the risk of obesity, diabetes, and cardiovascular complications [14]. Consequently, changes in sleep duration and sleep quality during pregnancy may adversely affect pregnancy outcomes, leading to intrauterine growth restriction, low birth weight, and preterm delivery [15,16]. Currently, there is insufficient research on the effects of maternal sleep on offspring growth since the research focus has been primarily on pregnancy outcomes. Therefore, the present study was conducted to examine the influence of multiple dimensions of the maternal sleep pattern on the occurrence of GDM and on the regulation of GDM effects on offspring physical growth.

## 2. Materials and Methods

### 2.1. Study Samples

A total of 6714 pregnant women were recruited in the ongoing Shanghai Maternal–Child Pairs Cohort (Shanghai MCPC, at two regional maternity hospitals in Shanghai, China from April 2016 to April 2018) [17]. In this study, 3329 eligible mother–child pairs were included, of which 382 pregnant women underwent objective sleep pattern measurements. Figure 1 details the participant selection process.

The sociodemographic and lifestyle factors and prenatal examination indicators of mother–child pairs were collected via questionnaire and medical records of the Maternal and Child Healthcare System. This cohort study followed the Strengthening the Reporting of Observational Studies in Epidemiology (STROBE) reporting guideline. Written informed consent was acquired from all of the study participants, and this cohort study was approved by the Ethics Committee of the School of Public Health, Fudan University (IRB number 2016-04-0587-EX).

### 2.2. Assessment of Sleep Pattern during Pregnancy

Sleep quality: The Chinese version of the Pittsburgh Sleep Quality Index (PSQI) was used to assess the sleep quality of pregnant women in recently months in both early and late pregnancy, and it has been validated for reliability and validity in pregnant women [18]. The total score of PSQI is 0–21. Higher total PSQI scores indicated worse sleep quality, with greater than five being considered as poor sleep quality and lower scores as good sleep quality [19]. In addition, we included the entry “Do you have the habit of napping” in the sleep pattern assessment to obtain information on napping in early and in late pregnancy.

Sleep duration: The night-time sleep duration in early and late pregnancy (16 and 34 weeks of gestation) was measured using a Xiaomi Bracelet wearable device, which uses various logic algorithms, meticulous scientific calculations, and a three-axis acceleration sensor (ADXL362) to identify and log the wearer’s activity. Devices were worn for one week for each trimester, and pregnant women were asked to send daily screenshots of all of the relevant data on sleep to the staff. To improve the data quality, all sleep durations <5 h or >9 h were additionally verified by a separate examiner. Night-time sleep duration groups were grouped according to sleep duration tertiles with cut-off levels of 7.5 and 8.5 h.

Changes in sleep quality during pregnancy (from early pregnancy to late pregnancy): These were classified into four mutually exclusive groups (Always good, Always poor, From good to poor, From poor to good) according to sleep quality as poor or good in early pregnancy and late pregnancy.

Changes in sleep duration during pregnancy: No case was found without a change in sleep duration in this study. From early pregnancy to late pregnancy, if the change in sleep duration was <0, it was defined as a shortened sleep duration during pregnancy; otherwise, it was defined as a prolonged sleep duration during pregnancy.

### 2.3. Diagnostic Criteria for GDM

In routine prenatal examinations, pregnant women underwent a 75 g oral glucose tolerance test at approximately 24 weeks gestation. The diagnosis of GDM was defined as one or more of three cut-off values (fasting plasma glucose: 5.1 mmol/L, 1-h glucose: 10.0 mmol/L, and 2-h glucose: 8.5 mmol/L) were met or exceeded [20].

### 2.4. Anthropometric Measures

Offspring anthropometric data were body length and body weight at birth and at eight time points for follow-up visits (at 1, 2, 4, 6, 9, 12, 18, and 24 months). Body mass index (BMI) was calculated by dividing body weight (kg) by height-squared (m^2^). The Z-score of growth was calculated using WHO Anthro version 3.2.2 (WHO, http://www.who.int/childgrowth/software/en/, accessed on 15 March 2022). The catch-up growth category and overweight/obese in early life classification were defined as two groups based on the BMI Z-score (BAZ): catch-up growth was a BAZ increment ≥0.67, and overweight/obese was defined as BAZ > 1 [21,22].

### 2.5. Covariates

Pre-pregnancy BMI was categorized into three groups: <18.5, 18.5–23.9, and ≥24.0 [23]. Maternal gestational weight gain was calculated as the difference between pre-pregnancy weight and weight before delivery. Maternal physical activity level (PA), anxiety, and depression during pregnancy were evaluated by professional scales or questionnaires (described previously) [24]. Maternal energy intake in late pregnancy was calculated by food exchange units using the information collected with the semi-quantitative Food Frequency Questionnaire [25]. Pregnancy complications were defined as preexisting conditions in pregnant women [26]. Data on maternal morning sickness in early pregnancy, feeding practices within the first 6 months, and breastfeeding duration were collected by questionnaires.

### 2.6. Statistical Analyses

Generalized linear model was used to analyze the effects of sleep patterns in early pregnancy and their change during pregnancy on the occurrence of GDM. We explored the association between gestational sleep patterns with the physical growth of offspring within 24 months by generalized linear mixed-effects models in both GDM and non-GDM group. Fixed effects were adjusted for pregnancy couple-related covariates, gestational weeks at delivery, sex, and feeding information, and no interactions were included. The random effects model incorporated within-individual repeated measurement error and between-individual random effects. We examined the distributions of sleep duration in early pregnancy and offspring overweight/obesity within 24 months by a smoothing plot, with an adjustment for potential covariates. Missing covariate data were imputed using multiple imputations (the number of imputed datasets was 20). Significance tests were two-tailed, and a *p*-value < 0.05 was considered to be statistically significant. All of the analyses were performed by using R software (version 4.0.5, R Foundation, Vienna, Austria).

## 3. Results

### 3.1. Participants

As shown in Table 1, pregnant women with GDM (*n* = 983, 23.85%) tended to be older at delivery (30.08 vs. 28.43 years, *p* < 0.001), had fewer gestational weeks at delivery (38.90 vs. 39.17 weeks, *p* < 0.001), gained less weight during pregnancy (13.78 vs. 15.09 kg, *p* < 0.001), and exhibited more pre-pregnancy overweight/obesity than women of the non-GDM group.

### 3.2. Distribution of Maternal Sleep Patterns and Its Effects on GDM

Pregnant women in the GDM group had higher total PSQI scores in early pregnancy (6.04 ± 2.90 vs. 5.70 ± 2.78) and lower scores in late pregnancy (6.45 ± 3.17 vs. 6.88 ± 3.24) than did the non-GDM group. The non-GDM group had a relatively higher percentage in group of always poor sleep quality from early to late pregnancy (non-GDM vs. GDM, 37.4% vs. 34.02%). The risk of GDM increased with an increasing total PSQI score in early pregnancy (OR = 1.03; 95% CI, 1.01 to 1.07). Compared to pregnant women with 7.5–8.5 h of sleep duration in early pregnancy, the risk of GDM increased 0.93-fold for those with ≥8.5 h of sleep (OR = 1.93; 95% CI, 1.01 to 3.81). Comparing changes of the maternal sleep pattern, the risk of GDM increased 0.48-fold when the sleep quality changed from poor to good from early to late pregnancy (OR = 1.48; 95% CI, 1.06 to 2.07) (Table 2). Additionally, in a separate analysis of all of the pregnant women with good sleep quality in late pregnancy, poor sleep quality in early pregnancy was found to remain a risk factor for GDM (OR = 1.46; 95% CI, 1.04 to 2.04).

### 3.3. Maternal Sleep Patterns on Offspring Growth

No significant effects were found for sleep patterns on birth outcomes, and the results of this analysis are not shown. For maternal sleep duration, in Figure 2, the risk of catch-up growth and overweight/obesity in offspring within 24 months was reduced by 45% and 29%, respectively (OR = 0.71; 95% CI, 0.51 to 0.99; OR = 0.71; 95% CI, 0.51 to 0.99) for each additional hour of maternal sleep duration in early pregnancy of the GDM group. Compared to pregnant women with 7.5–8.5 h of sleep duration in early pregnancy, those with <7.5 h or ≥8.5 h of sleep duration had an increased risk of overweight/obesity in offspring within 24 months by 1.73 and 1.43 times, respectively (OR = 2.73; 95% CI, 1.27 to 5.89; OR = 2.43; 95% CI, 1.04 to 5.69). For maternal sleep quality, pregnant women in the group of poor sleep quality from early to late pregnancy were associated with a 54% increased risk of overweight/obesity in offspring (OR = 1.54; 95% CI, 1.06 to 2.23), while an increased sleep duration from early to late pregnancy significantly reduced the risk of overweight/obesity by 76% (OR = 0.24; 95% CI, 0.07 to 0.76). For pregnant women in the non-GDM group, a change from poor to good sleep quality from early to late pregnancy was a protective factor for the occurrence of catch-up growth in offspring, reducing the risk by 20%.

Figure 3 demonstrated a U-shaped curve relationship between maternal sleep duration in early pregnancy and offspring overweight/obesity in early life, with almost 8 h as the cut-off point. Sensitivity analyses performed in the non-preterm birth population suggested that the primary findings were relatively robust (Appendix A, see Appendix A).

## 4. Discussion

This study found that sleep patterns during pregnancy affected the occurrence of GDM and were associated with offspring growth within 24 months. Sleep duration ≥8.5 h in early pregnancy or sleep quality changes from poor to good during pregnancy resulted in an increased risk of GDM. A U-shaped association existed between maternal sleep duration in early pregnancy and the risk of offspring overweight/obesity within 24 months, and 7.5–8.5 h was the optimal sleep duration. Poor gestational sleep quality also had a higher risk of offspring overweight/obesity. Maternal sleep quality changes from poor to good during pregnancy were associated with a reduced risk of offspring catch-up in the non-GDM group.

According to the aggregate data analysis, women with short sleep duration (6–7 h) were more likely to suffer from GDM (OR = 1.70) [11]. Moreover, pregnant women with ≤5 h of sleep duration were shown to have higher leptin levels than those with 7–8 h [27], and low leptin levels directly contributed to decreased insulin sensitivity, although the relationship between sleep duration and GDM were inconsistent [11]. The present study found that maternal sleep durations ≥8.5 h in early pregnancy had a greater risk of GDM, which was consistent with the results of two existing studies [28,29] that observed a U-shaped association between sleep duration during pregnancy and the risk of GDM. The different definitions of normal and abnormal sleep duration may lead to the different study results. Most studies define insufficient sleep as less than 7 h and excessive sleep as more than 9 h. Compared to other countries, pregnant women in China slept significantly less than 7 h per night [6], so 7.5 h was used as the threshold for insufficient sleep in the study.

Laboratory studies controlling for sleep quality in healthy adults have found that several days of sleep restriction were sufficient to cause a significant decrease in insulin sensitivity and impaired glucose tolerance [30]. Insulin resistance due to adaptive changes in glucose metabolism and pancreatic β-cell function in pregnant women begins to increase gradually in early and mid-pregnancy [31], and therefore, glucose metabolism in pregnant women during this period may be particularly vulnerable to insufficient sleep and poor sleep quality. A large cohort study concluded that pregnant women with poor sleep quality were more likely to develop GDM regardless of sleep duration [13]. Compared to women who slept well in early pregnancy, women with poorer sleep quality had a 77% increased risk of developing GDM later in life, while sleep quality in mid-pregnancy showed no association [13]. To date, evidence on the relationship between sleep quality and risk of GDM is limited and inconsistent. Elevated C-reactive protein may be a potential mechanism linking prediabetes and poor sleep quality that warrants further investigation [32]. Other potential mechanisms include enhanced oxidative stress, increased systemic inflammation, and a perturbed energy balance [33,34].

In a previous study, the effect of sleep duration during pregnancy on offspring development was focused on cesarean delivery and preterm delivery, and no association between sleep duration and small for gestational age or large for gestational age was found [35]. We found that maintaining a sleep duration of 7.5–8.5 h in early pregnancy or increasing the sleep duration from early to late pregnancy was associated with a lower risk of overweight/obesity in offspring within 24 months. Offspring of pregnant women with a prolonged sleep duration have been shown to have a lower mean birth weight [36], and offspring of those who slept more than 8 h in early pregnancy had a 39% lower risk of being overweight at five years of age [37]. Maternal sleep deprivation in late pregnancy was associated with higher BMI and waist circumference in children and increased risk of overweight/obesity [38]. These conflicting results may be due to differences in sociocultural background, inconsistencies in data collection methods and potential control variables, and different cut-off points for sleep duration.

This study found that, from early to late pregnancy, poor sleep quality among pregnant women with GDM was a risk factor for overweight/obesity, while sleep quality changing from poor to better during pregnancy reduced the risk of catch-up growth in offspring within 24 months among non-GDM pregnant women. Maternal sleep quality was generally poorer in late pregnancy compared to early pregnancy [26], and this resulted in lower birth weight in their offspring [39]. Pregnant women with poor sleep quality in late pregnancy may be more likely to have an unhealthy lifestyle, and the two factors can act together to affect fetal metabolic abnormalities and increase the risk of offspring overweight/obesity. The biological mechanism may be that maternal poor sleep quality leads to increased inflammation and may result in decreased bioavailability of nitric oxide, which in turn leads to fetal growth restriction. Short or poor sleep quality may lead to insulin resistance in pregnant women, resulting in a negative impact on birth weight [40]. Given the paucity of literature, the mechanisms by which sleep patterns in pregnant women with different GDM status led to differing physical growth of offspring in early life need to be explored, and the findings need to be validated by additional high-quality longitudinal studies.

Our study displayed several unique strengths. This study was based on a population-based prospective cohort to explore the impact on maternal glucose of sleep patterns in early pregnancy and on sleep pattern changes throughout pregnancy, and the impact of maternal sleep on offspring physical growth in early life under different GDM states. The sleep pattern throughout pregnancy was assessed from multiple dimensions (objective measurements to obtain sleep duration), compensating for the limitations of previous studies that focused on a single time point of sleep assessment. The confounding variables of the mother’s lifestyle (including napping) during pregnancy and the father’s demographic information were also controlled to ensure the stability of the results as much as possible. This study had some limitations. Bias existed in the subjective assessment of maternal sleep quality. Therefore, more dimensional and objective indicators are needed in the future to fully reflect the effects of maternal sleep quality. Information on medication use that may affect the maternal sleep pattern and on children’s energy intake that affects children’s physical growth was not collected, although maternal lifestyle and napping habits were controlled for to maintain stability of results. Furthermore, we combined offspring overweight and obesity for analysis (numerical limitations), but slight differences in the mechanisms of occurrence of the two may have confounded the understanding of the results. Ongoing follow-up information on children in Shanghai MCPC will be used to assess the long-term study results.

## 5. Conclusions

From the perspective of preventive health care, we found that maintaining a night-time sleep duration within 7.5–8.5 h in early pregnancy and ensuring good sleep quality throughout pregnancy were important for controlling maternal blood glucose and improving early life growth of offspring. These findings reinforced that giving targeted and practical sleep improvement advice to pregnant women with different blood glucose levels was important to control maternal blood glucose and optimize offspring growth during early life.

## Figures and Tables

**Figure 1 nutrients-14-03390-f001:**
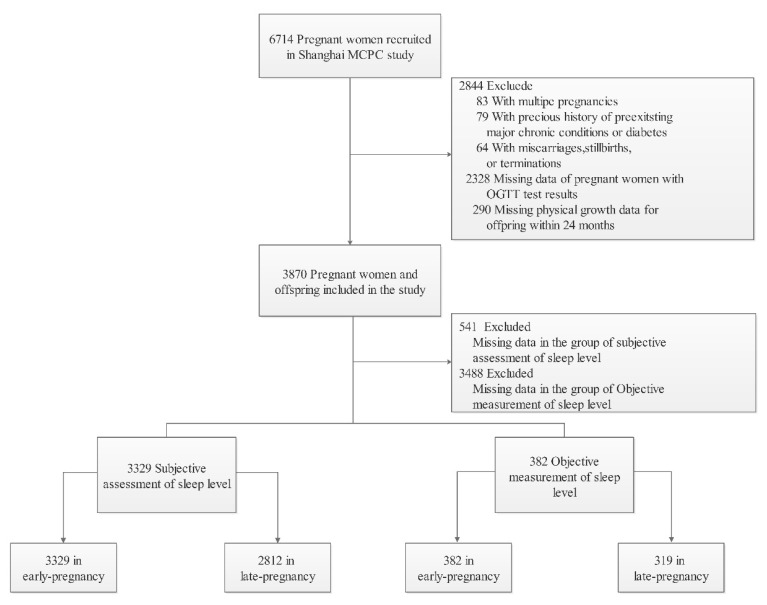
Flow diagram of study participants within the Shanghai Maternal–Child Pairs Cohort study.

**Figure 2 nutrients-14-03390-f002:**
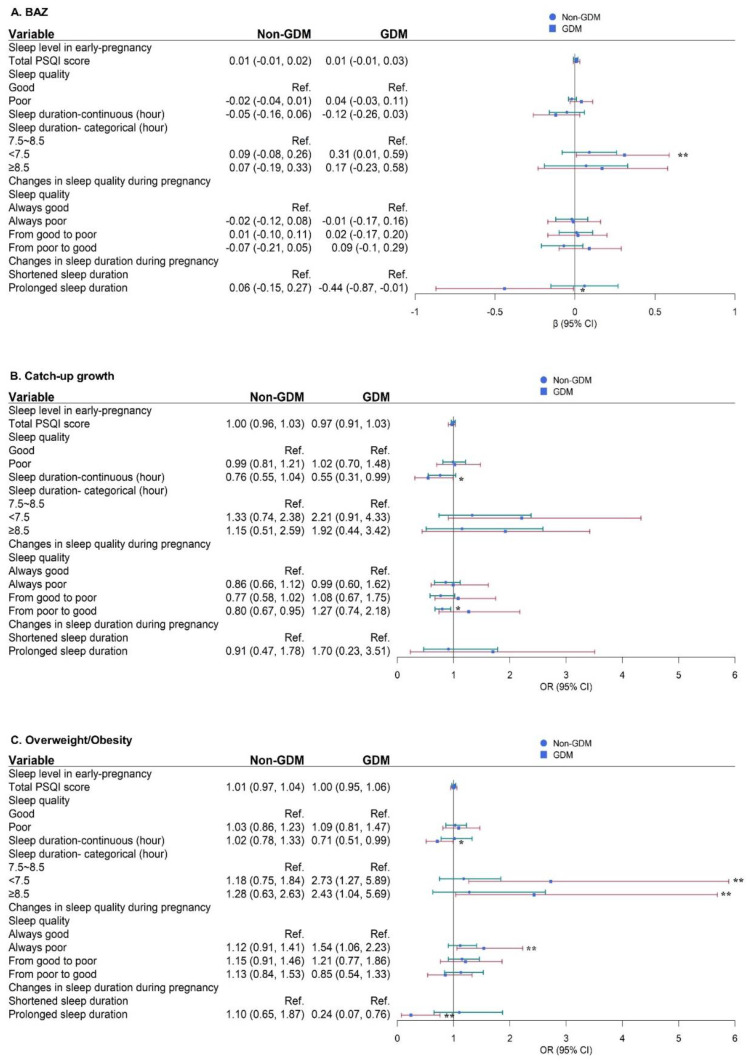
Effect of the maternal sleep pattern in early pregnancy and its change during pregnancy on physical growth of offspring within 24 months in the GDM subgroup [β/OR (95% CI)]. (**A**) Outcome is BAZ (BMI Z-score); (**B**) Outcome is catch-up growth; (**C**) Outcome is overweight/obesity; Green line: In the non-GDM population, the effect values for each group comparison; Red line: In the GDM population, the effect values for each group comparison; ** *p* < 0.01, * *p* < 0.05.

**Figure 3 nutrients-14-03390-f003:**
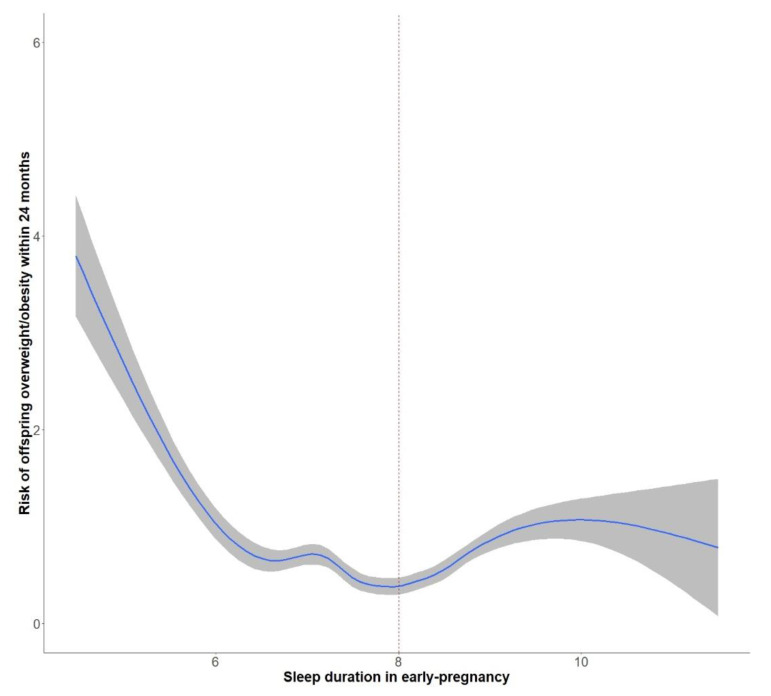
Association between maternal night−time sleep duration in early pregnancy and offspring growth in early life [OR (95% CI)]. Colored lines and shades around the line: OR with 95% CIs of effects of sleep duration in early pregnancy on offspring overweight/obesity within 24 months.

**Table 1 nutrients-14-03390-t001:** Characteristics of participants.

Variables	Participants, *n* (%).		
	Non-GDM (*n* = 2529)	GDM (*n* = 800)	*p* Value
Parents characteristics			
Maternal age, mean (SD), (Years)	28.43 (3.95)	30.08 (4.28)	<0.001
Gestational weight gain, mean (SD), (Kg)	15.09 (5.00)	13.78 (5.61)	<0.001
FPG in middle pregnancy, mean (SD), (mmol/L)	4.39 (0.31)	4.81 (0.59)	<0.001
1h plasma glucose level, mean (SD), (mmol/L)	7.12 (1.40)	9.49 (1.90)	<0.001
2h plasma glucose level, mean (SD), (mmol/L)	6.30 (1.06)	7.75 (1.73)	<0.001
Pre-pregnancy BMI (Kg/m^2^)			<0.001
18.5~23.9	1714 (67.77)	523 (65.38)	
<18.5	438 (17.32)	94 (11.75)	
≥24.0	377 (14.91)	183 (22.88)	
Education level > 12 (years)	2267 (89.64)	728 (91.00)	0.27
Total family income ≤ ¥200 thousand (RMB)	1868 (73.86)	574 (71.75)	0.13
Parity—primiparous	1469 (58.09)	409 (51.12)	<0.001
Depression in late pregnancy	312 (12.34)	97 (12.12)	0.87
Anxiety in late pregnancy	306 (12.10)	87 (10.88)	0.35
Energy intake in late-pregnancy, mean (SD), (kcal)	2254.62 (1182.63)	2020.78 (1072.19)	<0.001
Complications	769 (30.41)	141 (17.62)	0.16
PA level in late pregnancy—low	1271 (40.50)	375 (38.15)	0.41
Father’s age, mean (SD)	29.54 (4.53)	31.01 (5.01)	<0.001
Father’s BMI, mean (SD), (kg/m^2^)	23.82 (3.48)	23.98 (3.49)	0.26
Offspring characteristics			
Sex—male	1250 (49.43)	398 (49.75)	0.87
Gestational weeks at delivery, mean (SD),	39.17 (1.30)	38.90 (1.31)	<0.001
Delivery mode—Cesarean section	1294 (51.17)	432 (54.00)	0.16
Feeding practice within the first 6 months			
Exclusive breastfeeding	1141 (45.12)	333 (41.62)	0.17
Mixed feeding	796 (31.47)	277 (34.62)	
Bottle-feeding	592 (23.41)	190 (23.75)	
Breastfeeding duration, mean (SD), (months)	9.66 (5.09)	9.85 (5.07)	0.37

Data are shown as *n* (%) unless otherwise indicated. Based on χ^2^ test, with Fisher exact test used for variables with any cell count <10, or Kruskal–Walls test for continuous variables, *p* < 0.05.

**Table 2 nutrients-14-03390-t002:** Distribution of maternal sleep patterns in early pregnancy and their changes during pregnancy in the GDM subgroup and the risk of sleep patterns during pregnancy on GDM.

Variables	Participants, *n* (%)			OR (95% CI)
	Non-GDM	GDM	*p* Value	
Subjective Assessment (*n*1 = 2937, *n*2 = 933)				
Total PSQI score in early pregnancy, mean (SD)	5.70 (2.78)	6.04 (2.90)	0.003	1.03 (1.01, 1.07)
Sleep quality in early pregnancy			0.17	
Good	1278 (50.53)	382 (47.75)		Ref.
Poor	1251 (49.47)	418 (52.25)		1.09 (0.91, 1.31)
Changes in sleep quality during pregnancy			0.007	
Always good	472 (24.69)	166 (26.39)		Ref.
Always poor	715 (37.39)	214 (34.02)		0.81 (0.61, 1.07)
From good to poor	489 (25.58)	141 (22.42)		0.90 (0.67, 1.22)
From poor to good	236 (12.34)	108 (17.17)		1.48 (1.06, 2.07)
Objective measurement (*n*1 = 289, *n*2 = 93)				
Sleep duration in early pregnancy, continuous, (hour)	7.80 (0.81)	7.72 (1.07)	0.44	1.03 (0.77, 1.39)
Sleep duration in early pregnancy, categorical, mean (SD), (hour)			0.13	
7.5~8.5	144 (49.83)	37 (39.78)		Ref.
<7.5	97 (33.56)	33 (35.48)		1.09 (0.60, 1.99)
≥8.5	48 (16.61)	23 (24.73)		1.93 (1.01, 3.81)
Changes in sleep duration during pregnancy			0.28	
Shortened sleep duration	177 (73.75)	63 (79.75)		Ref.
Prolonged sleep duration	63 (26.25)	16 (20.25)		0.81 (0.41, 1.16)

*n*1 represented the non-GDM group, *n*2 represented the GDM group. Data are shown as *n* (%) unless otherwise indicated. Based on χ^2^ test, with Fisher exact test used for variables with any cell count <10, or Kruskal–Wallis test for continuous variables. Sleep pattern in early pregnancy: Adjusted for education, income, maternal age, pre-pregnancy BMI, complications, parity, morning sickness in early pregnancy, napping in early pregnancy, PA level in early pregnancy, history of diabetes, gestational weight gain, father’s BMI. Sleep pattern changes during pregnancy: Additionally adjusted for maternal smoking or drinking, depression, anxiety, PA level, energy intake, and napping in late pregnancy, gestational weeks at delivery.

## Data Availability

Not applicable.

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
