# Peer review of "Effects of Gestational Sleep Patterns and Their Changes on Maternal Glycemia and Offspring Physical Growth in Early Life"

_nutrients, 2022, doi:10.3390/nu14163390_

Round 1
Reviewer 1 Report
The very topic seems to of the publication seems to be extremely interesting. The authors present a study to examine the influence of multiple dimensions of the maternal sleep pattern on the occurrence of GDM and on the regulation of GDM effects on offspring physical growth. To date, little research has been done into the effects of maternal sleep patterns on the physical growth of offspring. All the more interesting is the fact that a very representative group of pregnant women was collected for the study. Additionally, the quality of the study is supported by a very representative group of pregnant women selected for the study.
Additionally, the manuscript is very well written with a clear and understandable English. The research seems to be designed appropriately.
1. Introduction
The introduction to the article defines the entire issue in a very clear and transparent manner therefore providing sufficient background and include all relevant references.
2. Materials and methods
The study appears to be properly designed while highlighting many variables. The methods are described in great detail and clearly.
Line 131-134: The sentence is too long and therefore a bit confusing.
3. Results
The way of presenting the results is very consistent and clear, while emphasizing the most important information.
4. Discussion
The discussion reveals the essence of the issue in a revealing way. Additionally, it contains very adequate references supporting the presented theses.
Author Response
Manuscript ID: nutrients-1870686
Tips: We thank all reviewers for their questions and suggestions about our manuscript. We make two revised versions: clean revised version and revised version using the "Track Changes" function. To make it easier for editors and reviewers to understand, the line number used in the response letter is from the clean revised version. The modified part of the clean revised version is highlighted in yellow color.
Reviewer1:
- Materials and methods
The study appears to be properly designed while highlighting many variables. The methods are described in great detail and clearly.
Line 131-134: The sentence is too long and therefore a bit confusing.
Response1: Thank you for your question. We are sorry that we did not describe our methods clearly. For clarity of expression, we split the complex sentence into two sentences “Generalized linear model was used to analyze the effects of sleep patterns in early pregnancy and their change during pregnancy on the occurrence of GDM. We explored the association between gestational sleep patterns with the physical growth of offspring within 24 months by generalized linear mixed-effects models in both GDM and non-GDM group.” (Page 4, lines 130-134).

Reviewer 2 Report
This study shows some interesting results to take into account
In future research. GDM is an important pathology in pregnants and its relation to sleep pattern during pregnancy ccould be a good perspective in clinic to improve pregnants health.
Author Response
Manuscript ID: nutrients-1870686
Tips: We thank all reviewers for their questions and suggestions about our manuscript. We make two revised versions: clean revised version and revised version using the "Track Changes" function. To make it easier for editors and reviewers to understand, the line number used in the response letter is from the clean revised version. The modified part of the clean revised version is highlighted in yellow color.
Reviewer2:
- In future research. GDM is an important pathology in pregnants and its relation to sleep pattern during pregnancy could be a good perspective in clinic to improve pregnants health.
Response1: Thanks for your kind comments. We strongly agree with your opinion. In the future, we will explore the long-term health effects of more dimensions of gestational sleep patterns on pregnant women or offspring in a larger prospective cohort.
